# AI literacy among basic school teachers in Ghana: A structural equation modelling analysis

Valentina Arkorful[1], Francis Arthur[2]*, Ernest Opoku[3], Ayishatu Ameen[4], Iddrisu Salifu[5], Sharon Abam Nortey[2], Solomon Adjatey Tetteh[2]

1 College of Distance Education, University of Cape Coast, Cape Coast, Ghana, 2 Department of Business and Social Sciences Education, Faculty of Humanities and Social Sciences Education, University of Cape Coast, Cape Coast, Ghana, 3 Institute of Education, School of Educational Development and Outreach, College of Education Studies, Ghana, 4 Department of Agricultural Economics and Extension, School of Agriculture, University of Cape Coast, Cape Coast, Ghana, 5 Centre for Coastal Management-Africa Centre of Excellence in Coastal Resilience, Department of Fisheries and Aquatic Sciences, University of Cape Coast, Cape Coast, Ghana

* a.francis1608@gmail.com, francis.arthur007@stu.ucc.edu.gh

## Abstract

Understanding the synergy of Artificial Intelligence (AI) literacy among teachers is imperative for successful integration in educational settings. This study examined basic school teachers' (BSTs) AI literacy based on four key aspects: AI knowledge and understanding (KUAI), AI application (AAI), AI application evaluation (EAIA), and AI ethics (AIE). The study employed a cross-sectional survey design and purposive sampling to collect data from 319 BSTs in Ghana. A variance-based structural equation modelling (VB-SEM) approach was used to explore the relationships between these dimensions and provide insights into their interrelationships. The study found that AAI positively influenced KUAI, EAIA, and AIE. However, AIE did not significantly influence KUAI and EAAI. In addition, KUAI did not influence EAIA. The findings contribute to understanding the current state of AI literacy among BSTs in Ghana and highlight areas for improvement in AI education and professional development programmes.

## 1. Introduction

The rapid advancement of artificial intelligence (AI) is transforming multiple sectors globally, with education emerging as a key domain of application. AI technologies are increasingly reshaping teaching and learning processes through personalised instruction, automation of administrative tasks, and enhanced student engagement [1–4]. As a result, education systems are under growing pressure to integrate AI meaningfully across all levels, including basic education.

While AI tools such as ChatGPT and Teacherbot offer opportunities to support instructional delivery and improve learning outcomes, their effective use depends largely on teachers' competencies [5–7]. In this context, AI literacy has become a

**Data availability statement:** The datasets generated and/or analysed during the current study are not publicly available due to ethical and legal restrictions imposed by the University of Cape Coast (UCC) Institutional Review Board (IRB), which approved the study protocol. The data contain potentially sensitive information from human participants, including responses that could indirectly identify individuals or reveal personal and professional characteristics of basic school teachers. Public deposition of such data would therefore violate the conditions under which ethical approval was granted, as well as participants' informed consent agreements guaranteeing confidentiality and anonymity. However, the data can be made available upon reasonable request for academic and non-commercial research purposes, subject to approval and in compliance with applicable ethical guidelines. Requests for data access should be directed to the Institutional Review Board (IRB), University of Cape Coast, via the official contact: irb@ucc.edu.gh. The IRB serves as an independent, non-author body responsible for reviewing and approving data access requests to ensure adherence to ethical standards and participant confidentiality.

**Funding:** The author(s) received no specific funding for this work.

**Competing interests:** The authors have declared that no competing interests exist.

critical requirement for educators. AI literacy encompasses not only knowledge and understanding of AI but also the ability to apply, evaluate, and ethically engage with AI technologies [8,9]. Without these competencies, the integration of AI into teaching may be ineffective or even problematic, particularly in relation to issues such as misinformation and ethical misuse [10,11].

Despite the increasing relevance of AI in education, evidence suggests that many teachers lack sufficient understanding of AI concepts and their pedagogical implications [12,13]. This challenge is more pronounced in developing contexts, where technological adoption is growing but often unsupported by adequate capacity building. In Ghana, although AI adoption in education is gradually expanding [14], limited attention has been given to assessing teachers' preparedness, particularly at the basic school level.

Existing studies in Ghana have largely focused on the adoption and use of AI tools, especially generative AI [15–17]. However, these studies provide limited insight into teachers' comprehensive AI literacy and are not specifically centred on basic school teachers. Furthermore, prior research has not consistently applied a structured theoretical framework that captures the multidimensional nature of AI literacy, such as the model proposed by Ng et al. [8], which integrates knowledge, application, evaluation, and ethics.

This gap highlights two key limitations in the literature: (1) the lack of empirical evidence on AI literacy among basic school teachers in Ghana, and (2) the absence of studies employing a comprehensive, theory-driven model to examine the interrelationships among AI literacy dimensions. Addressing these limitations is essential for informing policy, curriculum development, and teacher professional training.

Therefore, this study investigates AI literacy among basic school teachers in Ghana using the four-dimensional framework of Ng et al. [8] - knowledge and understanding of AI (KUAI), application of AI (AAI), evaluation of AI applications (EAI), and AI ethics (AIE). Unlike previous studies, this research adopts a variance-based structural equation modelling (VB-SEM) approach to examine the structural relationships among these dimensions. By doing so, the study provides a more rigorous and comprehensive understanding of AI literacy and contributes both theoretically and empirically to the field.

By fostering BST AI literacy and its implications for classroom practice, this research endeavours to empower teachers to navigate the AI-driven educational landscape effectively. This has long term impact, ultimately enhancing student learning outcomes and preparing the next generation for the challenges and opportunities of the digital age. Thus, this research has the potential to spur innovation and progress within the Ghanaian education sector by providing teachers with AI literacy. The rest of the paper is structured as follows: Section Two presents the hypotheses, while Section Three outlines the research methodology. Section Four presents the results, and Section Five is for the discussion. Finally, Section Six concludes the paper by emphasizing its theoretical significance, practical implications, limitations, and future prospects.

## 2. Literature review

### 2.1. Theoretical framework and hypotheses development

This study adopts the multidimensional conceptualisation of AI literacy proposed by Ng et al. [8,18], which extends beyond technical competence to include application, evaluation, and ethical engagement. Within this framework, AI literacy comprises four interrelated dimensions: knowledge and understanding of AI (KUAI), application of AI (AAI), evaluation of AI applications (EAI), and AI ethics (AIE). Although prior studies have defined AI literacy from different perspectives, there is convergence on its integrative nature, combining cognitive, practical, and ethical competencies [3,8,9]. Building on this foundation, the study examines the structural relationships among these dimensions in the context of BSTs.

### 2.2. Knowledge and understanding *of* AI and evaluation *of* AI application

Knowledge and understanding of AI (KUAI) constitute the foundation of AI literacy and shape teachers' ability to critically evaluate AI tools. Empirical evidence indicates that teachers with stronger AI knowledge demonstrate higher professional competence and more effective instructional practices [9]. Conceptually, AI literacy involves the ability to interpret and critically engage with AI systems [3], while also incorporating evaluative and ethical judgement [8]. These perspectives suggests that evaluation is inherently knowledge-driven. Without adequate understanding, BST are less able to assess the reliability, limitations, and pedagogical value of AI applications. Conversely, stronger KUAI enhances critical judgement and informed decision-making in classroom use. Therefore, we hypothesised that:

**H1:** "*Knowledge and understanding of AI has a significant positive influence on the evaluation of AI applications*".

### 2.3. Application *of* AI and knowledge and understanding *of* AI

The relationship between AI application (AAI) and knowledge is dynamic and reciprocal. Practical engagement with AI tools exposes teachers to real-world functionalities, thereby strengthening their conceptual understanding [9]. Evidence suggests that effective AI integration depends on teachers' ability to connect theoretical knowledge with practical use, particularly in designing meaningful learning experiences [19]. Through continued use, teachers refine their understanding of AI systems, including their capabilities and limitations. Thus, application serves as a mechanism for deepening knowledge and understanding. Therefore, we hypothesised that:

**H2a:** "*Application of AI has a significant positive influence on knowledge and understanding of AI*".

### 2.4. Application *of* AI and AI ethics

The application of AI in education introduces ethical concerns related to bias, transparency, and responsible use [4,20,21]. As teachers engage with AI tools, they become more aware of these issues and develop the capacity to apply ethical principles in practice. Ethical considerations are therefore embedded within the application process, guiding how AI is used in instructional contexts [22,23]. Empirical evidence further suggests that hands-on experience with AI enhances teachers' ability to recognise and address ethical implications [9,24]. Therefore, the researchers hypothesised that:

   H2b: "*AI application has a significant positive effect on AI ethics*".

### 2.5. Application *of* AI and evaluation *of* AI application

AI application extends beyond operational use to include critical assessment of AI tools. Teachers who actively engage with AI are better positioned to evaluate its effectiveness, limitations, and impact on learning outcomes [9,25]. Practical experience facilitates informed judgement, enabling teachers to assess alignment with pedagogical goals and ethical standards. This suggests that application strengthens evaluative capacity by providing experiential knowledge of AI systems. Therefore, we proposed the following hypothesis:

   H2c: "*The application of AI has a significant positive influence on evaluation of AI*".

 

### 2.6. AI ethics and evaluation *of* AI application

AI ethics (AIE) plays a critical role in shaping how teachers evaluate AI technologies. Ethical awareness enables teachers to assess AI tools in terms of fairness, accountability, and transparency, which are essential for responsible classroom integration [3,26]. Without ethical grounding, evaluation may overlook critical risks such as bias or misuse. Empirical studies indicate that ethical competence enhances teachers' ability to make informed and responsible decisions regarding AI adoption [9]. Hence, we postulated that:

H3$_a$: *"AI ethics has a significant positive influence on the evaluation of AI application".*

### 2.7. AI ethics and knowledge and understanding of AI

AI ethics contributes to deeper understanding by encouraging reflective engagement with AI technologies. Ethical considerations prompt teachers to examine how AI systems function and their broader societal implications, thereby strengthening conceptual knowledge [8,9]. This relationship highlights the integrative nature of AI literacy, where ethical awareness supports cognitive development. Teachers with stronger ethical understanding are therefore more likely to develop a comprehensive grasp of AI concepts and their applications. Consequently, this study also hypothesised that:

H$_{3b}$: *"AI ethics positively influences knowledge and understanding of AI".*

### 2.8. Conceptual model

The conceptual of model of this study is depicted by Fig 1.

## 3. Materials and methods

### 3.1. Study design and population

The study employed a descriptive cross-sectional survey design. This design is particularly effective for obtaining a snapshot of the current state of basic school teachers' artificial intelligence (AI) literacy at a specific point in time [27–29]. By collecting data at one point, the research aims to examine AI literacy among basic school teachers. This approach allows for the identification of patterns and correlations within the data, providing a foundation for further analytical exploration through variance-based structural equation modelling.

The population for this study comprised basic school teachers in Ghana. These educators are critical to the successful integration of AI literacy in the educational system, as they are directly responsible for implementing AI-related technologies and methodologies in the classroom. The study's target population included teachers from various schools, representing a diverse range of backgrounds and teaching experiences.

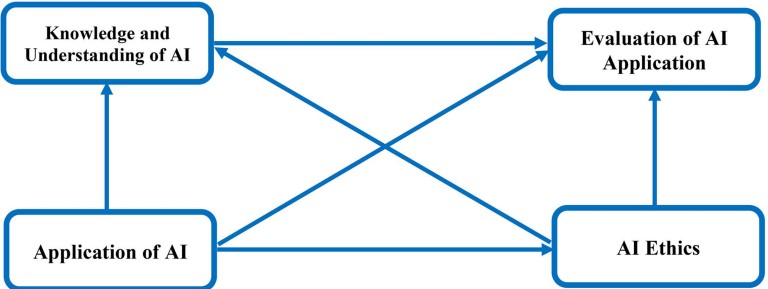

**Fig 1. Conceptual model.** Note: AI = Artificial intelligence. Source: Authors' construct.

Data were collected using a purposive sampling technique. Purposive sampling, a non-probabilistic method, was employed to gather the sample. This approach enabled the sample to be a deliberately selected subset of the population [30,31]. Teachers were purposively selected based on their prior awareness of AI, which, while ensuring informed responses, may introduce selection bias and limit the generalizability of the findings to the broader population of basic school teachers in Ghana who may have limited or no exposure to AI. A total of 319 basic school teachers, who were aware of AI, were chosen to participate in the study. The selected teachers were administered the AI literacy instrument. Teachers were given 15–25 minutes to complete the questionnaire, allowing ample time for thoughtful responses. Additionally, an informed consent form was provided to obtain their consent for participation, ensuring ethical standards were upheld. The data collection process was meticulously planned and executed to promote high response rates and enhance the reliability of the data collected.

### 3.2. Measure

This study builds on a refined set of criteria developed by Ng *et al.* [8] to measure the level of AI literacy of teachers. The original criteria were designed to assess the ability to use AI resources to support educational technology, but have been adapted to assess teachers' AI literacy in this study. The instrument contains a total of 20 items designed to collect information on four different dimensions of AI literacy: Knowing and Understanding AI (KUAI), Applying AI (AAI), Evaluating AI Application (EAIA), and AI Ethics (AIE). Each item is measured on a 5-point Likert scale, ranging from 1 (no AI literacy at all) to 5 (complete dominance of AI literacy).

### 3.3. Data analysis

To analyse the survey data, a variance-based (VB) structural equation modelling (SEM) approach was employed This analytical technique is ideal for studies exploring new areas, such as this one, because it allows us to explore complex relationships between data that we can measure directly (observed variables) and underlying concepts we cannot measure directly (latent variables) [32,33]. Using VB-SEM, the researchers assessed how well the instrument measured AI literacy (validity) and its consistency (reliability). We also explored how the different dimensions of AI literacy (such as understanding and applying AI) related to each other.

### 3.4. Ethical considerations

The study prioritised the ethical treatment of participants. Ethical approval for the study was granted by the Institutional Review Board of the University of Cape Coast (ID: UCCIRB/CES/2023/169). Following the approval, data collection took place from March 04, 2024, to July 26, 2024. All participants were made to sign written informed consent before taking part, and the researchers guaranteed the confidentiality and anonymity of their responses. In accordance with ethical guidelines, participants were free to withdraw from the study at any time without consequence. To ensure data security and privacy, information was stored securely and only the research team had access to it.

## 4. Results

### 4.1. Measurement model assessment

To make sure the assessment tool is reliable and accurate, we checked how consistent and valid the variables are. Table 1 shows the detailed results for these checks, including factor loadings, different measures of reliability, and measures of variance for each construct: Cronbach's alpha (α), Rho_A, composite reliability (CR), average variance extracted (AVE), and inner variance inflation factor (VIF)

### 4.2. Indicator loadings and construct reliability

All items had factor loadings above the threshold of 0.70, signifying that each item contributed significantly to its respective construct. The loadings ranged from 0.814 to 0.948 across the constructs (see Table 1 and S1 Table), indicating

**Table 1. Construct Reliability and Validity.**

| Constructs | Items | Loadings | α | Rho_A | CR | AVE | p value |
|---|---|---|---|---|---|---|---|
| AAI | AA1 | 0.928 | 0.921 | 0.921 | 0.950 | 0.864 | *** |
|  | AA2 | 0.948 |  |  |  |  | *** |
|  | AA3 | 0.911 |  |  |  |  | *** |
| AIE | AE1 | 0.879 | 0.940 | 0.946 | 0.954 | 0.808 | *** |
|  | AE2 | 0.927 |  |  |  |  | *** |
|  | AE3 | 0.941 |  |  |  |  | *** |
|  | AE4 | 0.926 |  |  |  |  | *** |
|  | AE5 | 0.814 |  |  |  |  | *** |
| EAIA | EAA1 | 0.936 | 0.968 | 0.968 | 0.975 | 0.886 | *** |
|  | EAA2 | 0.936 |  |  |  |  | *** |
|  | EAA3 | 0.946 |  |  |  |  | *** |
|  | EAA4 | 0.947 |  |  |  |  | *** |
|  | EAA5 | 0.939 |  |  |  |  | *** |
| KUAI | KUA1 | 0.920 | 0.930 | 0.936 | 0.955 | 0.876 | *** |
|  | KUA2 | 0.955 |  |  |  |  | *** |
|  | KUA3 | 0.934 |  |  |  |  | *** |

Note: AAI = Application of AI; AIE = AI ethics; EAIA = Evaluation of AI application; KUAI = Knowledge and understanding of AI; α = Cronbach's Alpha; CR = Composite Reliability; AVE = Average Variance Extracted.

strong correlations between items and their underlying factors. Also, reliability of the constructs was assessed using Cronbach's alpha (α), composite reliability (CR), and Rho_A. All constructs demonstrated high reliability, with Cronbach's alpha values well above the acceptable threshold of 0.70, indicating consistent internal reliability. Specifically, the values for Cronbach's alpha were 0.921 for Applying AI (AAI), 0.940 for AI Ethics (AIE), 0.968 for Evaluating AI Application (EAIA), and 0.930 for Knowing and Understanding AI (KUAI) [see Table 1].

Additionally, the composite reliability (CR) values for all constructs exceeded the recommended threshold of 0.70, confirming the internal consistency of the constructs [34]. The CR values were 0.950 for AAI, 0.954 for AIE, 0.975 for EAIA, and 0.955 for KUAI. Additionally, Rho_A values, which provide an alternative measure of reliability, were also high, corroborating the reliability of the constructs.

## 4.3. Convergent validity

Construct validity was assessed through average variance extracted (AVE). The AVE values for each construct were well above the minimum acceptable level of 0.50, which indicates a good level of convergent validity [34]. The AVE values were 0.864 for AAI, 0.808 for AIE, 0.886 for EAIA, and 0.876 for KUAI (see Table 1). These high AVE values suggest that the constructs explain a significant portion of the variance in their respective items. The results of the PLS-SEM algorithm are displayed by Fig 2.

## 4.4. Multicollinearity

The analysis of multicollinearity, using variance inflation factor (VIF) values, reveals that multicollinearity is not a significant concern within the measurement model. The VIF values for the constructs - Applying AI (AAI), AI Ethics (AIE), Evaluating AI Application (EAIA), and Knowing and Understanding AI (KUAI) – are all below the critical threshold of 5, indicating that the constructs are sufficiently independent [35]. Specifically, the VIF values range from 1.000 to 3.491 (see Table 2), suggesting

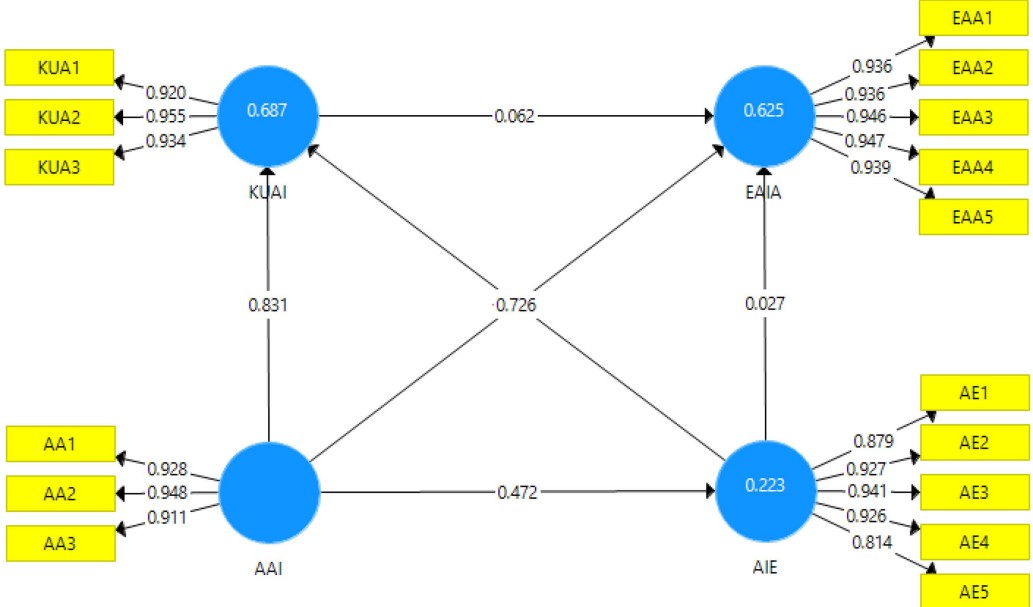

**Fig 2. PLS-SEM algorithm results.**

**Table 2. Results for the VIF.**

|       | AAI   | AIE   | EAIA  | KUAI  |
|-------|-------|-------|-------|-------|
| AAI   |       | 1.000 | 3.491 | 1.286 |
| AIE   |       |       | 1.286 | 1.286 |
| EAIA  |       |       |       |       |
| KUAI  |       |       | 3.190 |       |

Note: AAI = Application of AI; AIE = AI ethics; EAIA = Evaluation of AI application; KUAI = Knowledge and understanding of AI

that the variables are not highly correlated. Consequently, the model's robustness and the validity of the findings in assessing basic school teachers' AI literacy in Ghana are supported, as multicollinearity does not significantly undermine the estimation of the structural equation model.

## 4.5. Discriminant validity

We tested how well the measures discriminated between different concepts using two methods: the "Fornell-Larcker criterion and the Heterotrait-Monotrait Ratio (HTMT)". The Fornell-Larcker criterion suggests that a measure has good discriminant validity if the square root of its average variance extracted (AVE) is higher than its correlation with any other measure. The diagonal values, representing the square roots of the AVE, are 0.929 for AAI, 0.899 for AIE, 0.941 for EAIA, and 0.936 for KUAI, all of which are greater than their corresponding inter-construct correlations [34] (see Table 3). Additionally, the HTMT values, which should ideally be below 0.90 [36], are 0.506 between AAI and AIE, 0.835 between AAI and EAIA, 0.411 between AIE and EAIA, 0.893 between AAI and KUAI, 0.411 between AIE and KUAI, and 0.705 between

**Table 3. Fornell-Larcker criterion.**

| Constructs | AAI | AIE | EAIA | KUAI |
|---|---|---|---|---|
| AAI | **0.929** | | | |
| AIE | 0.472 | **0.899** | | |
| EAIA | 0.790 | 0.393 | **0.941** | |
| KUAI | 0.829 | 0.386 | 0.673 | **0.936** |

Note: AAI = Application of AI; AIE = AI ethics; EAIA = Evaluation of AI application; KUAI = Knowledge and understanding of AI

**Table 4. Heterotrait-Monotrait Ratio (HTMT).**

| | AAI | AIE | EAIA | KUAI |
|---|---|---|---|---|
| AAI | | | | |
| AIE | 0.506 (0.402; 0.605) | | | |
| EAIA | 0.835 (0.769; 0.892) | 0.411 (0.305; 0.509) | | |
| KUAI | 0.893 (0.832; 0.941) | 0.411 (0.301; 0.514) | 0.705 (0.621; 0.776) | |

Note: AAI = Application of AI; AIE = AI ethics; EAIA = Evaluation of AI application; KUAI = Knowledge and understanding of AI

EAIA and KUAI (see Table 4). These values are within acceptable limits, thereby supporting the discriminant validity of the constructs in the model, indicating that the constructs are distinct and measure different concepts in the context of basic school teachers' AI literacy in Ghana.

## 4.6. Model Fit Assessment

The model fit assessment revealed comparable fit indices between the saturated and estimated models, indicating a satisfactory fit of the estimated model to the data. The Standardised Root Mean Square Residual (SRMR) remained consistent at 0.052 for both models, suggesting minimal discrepancies between observed and predicted covariance matrices. Additionally, the goodness-of-fit indices, d_ULS and d_G, exhibited identical values of 0.368, indicating a close fit between the estimated model and the data. The Chi-Square statistic, though large in absolute terms, was consistent across both models at 1063.124, which is expected given the sensitivity of Chi-Square to sample size. Furthermore, the Normed Fit Index (NFI) yielded a value of 0.839 for both the saturated and estimated models, indicating a moderate level of fit. Overall, the model fit indices collectively suggest that the estimated model adequately captures the underlying relationships among the variables, supporting its validity for further analysis and interpretation. Table 5 shows the results of the model fit indices from the PLS-SEM algorithm.

## 4.7. Evaluation of structural model

We looked at how the different concepts in the model related to each other. This helped us to understand the direct effects of these relationships. We also assessed how well the model explained the data and the strength of these effects. The analysis focused on path coefficients (β), which indicate the direction and strength of the relationships, and their significance levels. These levels of significance are indicated by t-values and p-values.

The relationship between knowledge and understanding of AI (KUAI) and evaluating AI application (EAIA) is not significant ($\beta = 0.062$, $t = 0.771$, $p = 0.441$) [see Table 6]. In contrast, Applying AI (AAI) has a strong and significant effect on KUAI ($\beta = 0.831$, $t = 34.435$, $p < 0.001$), AI Ethics (AIE) ($\beta = 0.472$, $t = 9.814$, $p < 0.001$), and EAIA ($\beta = 0.726$, $t = 9.111$, $p < 0.001$).

**Table 5. Model Fit Indices.**

| Fit Indices | Saturated Model | Estimated Model |
|---|---|---|
| SRMR | 0.052 | 0.052 |
| d_ULS | 0.368 | 0.368 |
| d_G | 0.584 | 0.584 |
| Chi-Square | 1063.124 | 1063.124 |
| NFI | 0.839 | 0.839 |

Note: AAI = Application of AI; AIE = AI ethics; EAIA = Evaluation of AI application; KUAI = Knowledge and understanding of AI

**Table 6. Structural model.**

| Hypotheses | β | Sample Mean (M) | SD | T value | p values | 2.5% | 97.5% |
|---|---|---|---|---|---|---|---|
| KUAI -> EAIA | 0.062 | 0.058 | 0.080 | 0.771 | 0.441 | −0.100 | 0.211 |
| AAI -> KUAI | 0.831 | 0.831 | 0.024 | 34.435 | <.001 | 0.781 | 0.875 |
| AAI -> AIE | 0.472 | 0.474 | 0.048 | 9.814 | <.001 | 0.373 | 0.562 |
| AAI -> EAIA | 0.726 | 0.729 | 0.080 | 9.111 | <.001 | 0.566 | 0.875 |
| AIE -> EAIA | 0.027 | 0.027 | 0.051 | 0.532 | 0.595 | −0.075 | 0.125 |
| AIE -> KUAI | −0.006 | −0.005 | 0.028 | 0.201 | 0.840 | −0.063 | 0.049 |

Note: "SD = Standard Deviation, AAI = Application of AI; AIE = AI ethics; EAIA = Evaluation of AI application; KUAI = Knowledge and understanding of AI"

However, the paths from AIE to EAIA (β = 0.027, t = 0.532, p = 0.595) and from AIE to KUAI (β = −0.006, t = 0.201, p = 0.840) are not significant, indicating that AIE does not significantly influence these constructs (see Table 6). While the model demonstrates several strong and significant relationships, the presence of non-significant paths (e.g., KUAI -> EAIA and AIE -> EAIA/KUAI) suggests that not all proposed relationships are supported. Fig 3 shows the bootstrapping results from PLS-SEM.

## 4.8. Explanatory power and effect size of the model

The R-square ($R^2$) values indicate the proportion of variance explained by the predictor variables for each endogenous construct. The $R^2$ values are 0.223 for AIE, 0.625 for EAIA, and 0.687 for KUAI, with adjusted $R^2$ values being 0.220, 0.622, and 0.685, respectively [see Table 7]. These values suggest that the model explains a substantial portion of the variance in EAIA and KUAI, while the variance explained in AIE is weak [34]). In addition, the f-square ($f^2$) values assess the impact of each predictor construct on the dependent constructs. AAI has a substantial effect on KUAI ($f^2$ = 1.714) and EAIA ($f^2$ = 0.403) [37,38], and medium effect on AIE ($f^2$ = 0.286) [see Table 8]. Conversely, AIE has negligible effects on EAIA ($f^2$ = 0.002) and KUAI ($f^2$ = 0.000), indicating that AAI is the primary driver in the model. These results highlight the critical role of AAI in influencing other constructs within the structural model, reinforcing its importance in enhancing basic school teachers' AI literacy in Ghana.

## 4.9. Revised conceptual model

The revised conceptual of model of this study is depicted by Fig 4.

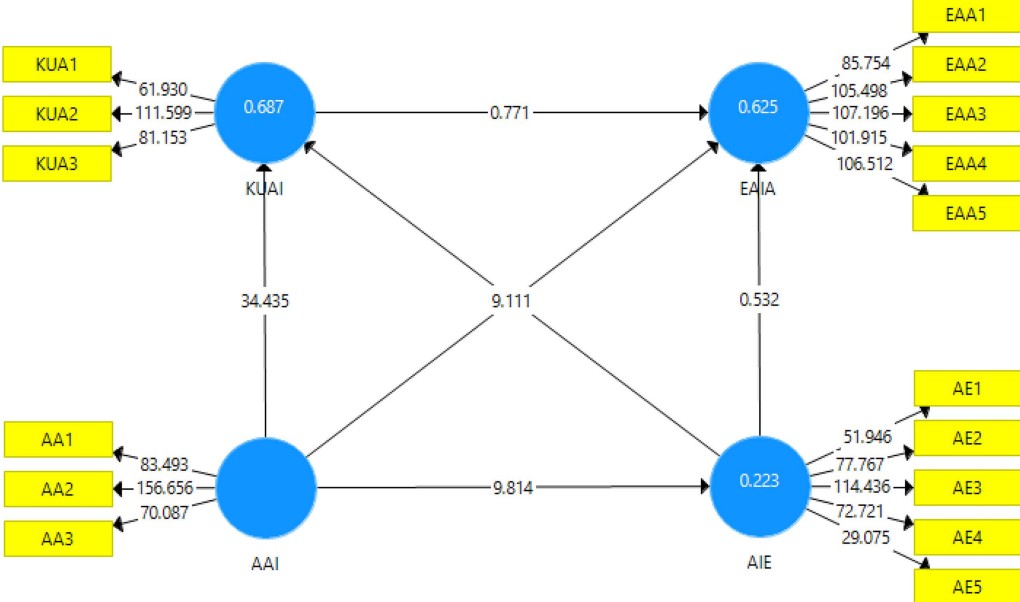

**Fig 3. Bootstrapping results from PLS-SEM.**

**Table 7. Coefficient of determination (*R²*).**

| Constructs | R Square | R Square Adjusted |
|---|---|---|
| AIE | 0.223 | 0.220 |
| EAIA | 0.625 | 0.622 |
| KUAI | 0.687 | 0.685 |

Note: AAI = Application of AI; AIE = AI ethics; EAIA = Evaluation of AI application; KUAI = Knowledge and understanding of AI

## 5. Discussion

The use of transformative technologies has become a fundamental aspect of professional development at various sectors and the education sector is no exception. The transformative potential of AI for shaping teaching and learning is undeniable. As AI advances at a rapid pace, it is crucial to understand the multifaceted nature of AI literacy among teachers. Inspired by this, we explored the core dimension of AI literacy, including knowledge and understanding of AI (KUAI), applications of AI (AAI), evaluation of AI applications (EAIA), and ethics of AI (AIE), needed for responsible implementation by primary school teachers. This study used 319 basic school teachers and employed a variance-based structural equation modelling (VB-SEM) to test the relationships between the dimensions of AI literacy. The aim of the study was to identify the factors that influence basic school teachers' AI literacy. A total of six hypotheses were tested, of which the results validated three hypotheses and did not support three.

The results of the study showed no significant influence of KUAI on EAIA, suggesting that knowledge and understanding of AI does not significantly influence the evaluation of AI by basic school teachers. This may be because most of these basic school teachers rely on preconceived notions or external opinions, leading to bias or overconfidence. In addition, basic school teachers' theoretical knowledge and understanding of AI may not align with the practical outcomes

**Table 8. Effect size (F-square [$f^2$]).**

|  | AAI | AIE | EAIA | KUAI |
|---|---|---|---|---|
| AAI |  | 0.286 | 0.403 | 1.714 |
| AIE |  |  | 0.002 | 0.000 |
| EAIA |  |  |  |  |
| KUAI |  |  | 0.003 |  |

Note: AAI = Application of AI; AIE = AI ethics; EAIA = Evaluation of AI application; KUAI = Knowledge and understanding of AI

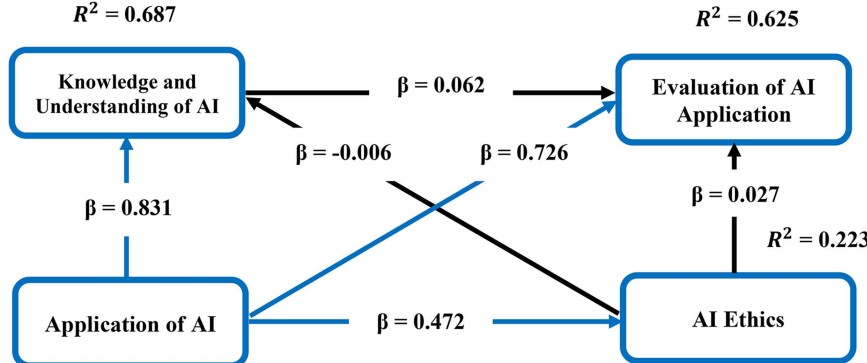

**Fig 4. Revised conceptual model.** Note: → Non-significant path. Source: Authors' construct.

and real-world effectiveness of AI applications, making their actual AI literacy a less important factor in the evaluation of AI applications. This finding contradicts previous research that emphasises that teachers' knowledge and understanding of AI has closely linked with the evaluation of AI applications [9]. Therefore, there is an urgent need to improve the AI literacy of Ghanaian basic school teachers, both in terms of knowledge and understanding of AI (KUAI) and evaluation of AI applications (EAIA). To address this, practical policy implications should include integrating comprehensive AI training programmes into teacher education, providing ongoing professional learning communities (PLCs) focused on AI, and ensuring access to AI resources. This will enable basic school teachers to develop technical knowledge and understanding of AI, leading to the evaluation of AI applications and more informed and effective use of AI in the classroom.

In addition, this study found a significant relationship between AAI and KUAI. This highlights that the use of AI by primary school teachers has a significant positive impact on improving their knowledge and understanding of AI, which is a core dimension of AI literacy. This finding is expected and can be attributed to the common notion that practical, hands-on experience in the iterative learning process with increased engagement and motivation are factors that contribute to deepening technological literacy. In this context, basic school teachers can better develop AI literacy in terms of knowledge and understanding to support their teaching practice and improve the overall quality of education if they continue to apply AI tools. The findings of this study are consistent with those of previous research emphasising the importance of teachers' application of AI in determining the knowledge and understanding of AI in their professional practice [19]. Therefore, it is recommended that basic school teachers' knowledge and understanding of AI be further enhanced by encouraging their use of AI tools. This can be achieved by integrating AI-based tools into teacher training, promoting practical AI projects, and providing continuous AI-focused professional development.

This study shows that AI literacy has a significant positive impact on AI ethics among primary school teachers in Ghana. Thus, through the application of AI, teachers develop AI literacy on how to articulate the ethical aspects of AI

responsibly in educational settings and share these ideas with their peers and students. This finding supports those of previous studies [9,24] and emphasises the importance of AI literacy. The positive influence of application of AI on AI ethics among Ghanaian primary school teachers can be attributed to increased AI literacy. As teachers become more knowledgeable about AI, they are better equipped to understand ethical considerations and use AI responsibly. This leads to effective and ethical implementation of AI tools in the classroom, promoting a culture of responsible AI use and improving educational outcomes. To ensure that teachers are well prepared to use AI responsibly, it is recommended that AI literacy is integrated into teacher training programmes, that continuous professional development is provided, and that ethical guidelines for the use of AI in education are established.

More importantly, the results showed that AAI has a significant positive impact on EAI. This emphasises that the application of AI by basic school teachers can positively enhance their AI literacy in terms of evaluating the application of AI, which is consistent with the findings of previous studies [9,25]. Thus, as teachers integrate AI tools into their teaching practice and continuously apply or use them, their understanding and evaluation of AI technology also improves. Teachers' improved AI literacy is likely due to hands-on experience, which enhances their ability to critically evaluate and effectively use AI in educational contexts. Given that the use of AI has a significant positive impact on teachers' AI literacy in terms of evaluating such AI tools, there is a need for policies that promote the integration of AI into teacher training programmes. Basic school teachers should be given hands-on experience with AI to improve their literacy and evaluation skills. Policies should focus on promoting the integration of AI in primary education and provide teachers with continuous training in the use of these technologies. This will enable basic teachers to make effective use of AI, improve their teaching methods and ensure that pupils benefit from advanced technology in the classroom. This policy approach supports sustainable improvements in teaching quality and AI literacy.

The literature suggests that teachers' AI literacy in AI ethics is essential for fostering positive attitudes, knowledge and skills relevant to their future studies and careers. Therefore, one of the questions we wanted to address was whether AI ethics determines teachers' evaluation of AI applications as AI literacy. However, the results showed that AIE had no significant effect on EAI. Thus, our findings suggest that primary school teachers' understanding of AI ethics does not directly affect their ability to evaluate AI applications. While training in AI ethics is crucial, it does not necessarily improve practical evaluation skills. This seems to contradict the findings that knowledge of AI ethics significantly improves the evaluation of AI applications [8,9,26]. To address this gap, it is therefore necessary to integrate targeted technical training alongside AI ethics in professional development programmes. We recommended that policymakers emphasize hands-on experience with AI tools to improve practical evaluation skills. This would enable primary school teachers to balance knowledge of AI ethics with practical evaluation of applications to ensure comprehensive AI literacy, thereby improving their overall effectiveness in using AI in education.

Furthermore, the literature on AI literacy suggests that AI ethics and knowledge and understanding of AI go hand in hand. However, the results of our study showed that AIE did not significantly influence KUAI. Thus, AI ethics, conceptualised as primary teachers' understanding of the responsible use of AI, does not have a significant effect on knowledge and understanding of AI applications. This finding implies that ethics training alone does not improve teachers' knowledge and understanding of AI applications. However, this contradicts the finding that teachers' AI ethical literacy in turn enhances their knowledge and understanding of AI [9,36]. To improve teachers' knowledge and understanding of AI, integrate practical AI training with ethics education. Provide hands-on workshops and technical courses focused on AI applications. This combined approach will ensure that teachers gain both ethical insights and practical skills, leading to a well-rounded AI literacy that is essential for effective teaching.

## 6. Conclusion

This paper aims to add to the limited evidence on teachers' AI literacy, particularly from primary school teachers who play a critical role in shaping students' foundations. This study proposes a model that combines key dimensions of AI literacy, such as knowledge and understanding of AI, applications of AI, evaluation of AI applications, and ethics of AI. The

results show a significant relationship between application of AI and knowledge and understanding of AI, highlighting the importance of encouraging teachers to use AI tools to improve their literacy in terms of knowledge and understanding. Hands-on AI projects and continuous professional development are recommended to deepen technological literacy and improve the quality of teaching. The study also shows that application of AI has a positive impact on evaluation of AI application, suggesting that hands-on AI experience improves teachers' ability to effectively evaluate AI applications.

Conversely, there is no significant effect of knowledge and understanding of AI on evaluation of AI applications, suggesting that theoretical knowledge of AI does not translate into practical evaluation skills, often due to reliance on external opinions and biases. To improve AI literacy, comprehensive AI training programmes should be integrated into teacher education, with a focus on practical, hands-on experience. Similarly, AI ethics showed no significant effect on the evaluation of AI applications or knowledge and understanding of AI, suggesting that ethics training alone is not sufficient to improve practical AI literacy. This calls for a combined approach of ethical and technical training to ensure comprehensive AI literacy among primary school teachers in Ghana. Policymakers should promote the integration of AI into teacher training, provide continuous AI-focused development, and establish ethical guidelines to promote the responsible use of AI in education, ultimately improving educational outcomes and the quality of teaching.

## 7. Contribution and implications of the study

This study provides crucial insights into the integration of Artificial Intelligence (AI) literacy among basic school teachers (BSTs), a critical area of inquiry as educational systems globally continue to embrace technological advancements. While previous studies have primarily focused on general digital literacy and technological pedagogical content knowledge (TPACK), this study moves beyond foundational digital skills by delving into the nuanced aspects of AI literacy specifically. AI is rapidly becoming a vital component of the educational technology landscape, and understanding how teachers interact with this emerging technology is crucial for effective integration in classrooms.

One significant contribution of this study lies in its multidimensional approach to AI literacy, assessing not only teachers' knowledge and understanding of AI but also their capacity to apply AI tools, evaluate AI applications, and navigate ethical issues related to AI in educational settings. By exploring these four key dimensions – AI knowledge and understanding (KUAI), AI application (AAI), AI application evaluation (EAIA), and AI ethics (AIE) – this study advances the conversation on teacher preparedness in integrating AI in teaching and learning processes.

Furthermore, the use of variance-based structural equation modelling (VB-SEM) allows for a sophisticated examination of the interrelationships between these dimensions, offering novel insights into how various aspects of AI literacy are interconnected. For instance, the finding that AI application (AAI) significantly influences teachers' knowledge, evaluation, and ethics in AI underscores the importance of practical AI exposure and training for teachers. This insight suggests that hands-on experience with AI technologies may be a critical factor in fostering broader AI literacy, which has implications for teacher education and professional development programs.

The study's contribution extends beyond AI literacy by informing the design and implementation of teacher training programs focused on AI. With the understanding that AI application drives other dimensions of AI literacy, teacher education institutions can prioritise experiential learning opportunities and collaborative AI-based projects to deepen teachers' competency. Moreover, the emphasis on AI ethics highlights the need for teacher education programmes to incorporate critical discussions about the moral and societal implications of AI use in schools, ensuring that teachers can responsibly integrate AI technologies.

In summary, this study contributes significantly to the field of technology integration for teacher education by providing an empirical basis for understanding AI literacy among BSTs in Ghana. The findings support the development of more targeted AI-focused teacher education initiatives, positioning AI as a pivotal area for technological innovation in education. The research not only addresses current gaps in the literature but also sets a foundation for future studies to explore how AI literacy can be effectively cultivated and sustained among educators globally.

## 8. Strengths and limitation of the study

To the best of our knowledge, this study is the first to explore the relationship between the dimensions of AI literacy of primary school teachers in Ghana. Using a variance-based structural equation modelling (VB-SEM) approach, this study provides new insights into AI literacy among basic school teachers in Ghana based on four key aspects: knowledge and understanding of AI, application of AI, evaluation of AI applications, and AI ethics, with important implications for policy and practice. Nevertheless, some limitations for future research need to be acknowledged. In particular, the cross-sectional descriptive design of the study limits the ability to establish causal relationships between the variables examined. Therefore, this design lacks the ability to track changes over time. In contrast, longitudinal studies can follow participants over time, allowing researchers to observe how changes in one variable precede changes in another, thus facilitating the assessment of causal patterns. Second, the current research focused exclusively on basic school teachers in Ghana. Future research involving teachers at higher levels of education or from a different country would strengthen the general-isability of the study's conclusions. The self-reported nature of the data introduces the possibility of bias. Future research could explore alternative methods such as experimental designs and mixed methods approach. Likewise, future studies should focus on the use of advanced AI models such as artificial neural networks (ANNs), which have become increasingly common in recent studies to enhance prediction and support PLS-SEM findings, particularly when dealing with multidimensional data, as in AI-assisted literacy research.

## Supporting information

**S1 Table. Cross loading.**
(PDF)

## Author contributions

**Conceptualization:** Valentina Arkorful, Francis Arthur, Ernest Opoku, Ayishatu Ameen, Iddrisu Salifu, Sharon Abam Nortey, Solomon Adjatey Tetteh.

**Data curation:** Francis Arthur.

**Formal analysis:** Francis Arthur, Iddrisu Salifu.

**Methodology:** Francis Arthur, Iddrisu Salifu.

**Software:** Francis Arthur.

**Supervision:** Ernest Opoku.

**Validation:** Francis Arthur, Iddrisu Salifu.

**Visualization:** Francis Arthur.

**Writing – original draft:** Valentina Arkorful, Francis Arthur, Ernest Opoku, Ayishatu Ameen, Iddrisu Salifu, Sharon Abam Nortey, Solomon Adjatey Tetteh.

**Writing – review & editing:** Valentina Arkorful, Francis Arthur, Ernest Opoku, Ayishatu Ameen, Iddrisu Salifu, Sharon Abam Nortey, Solomon Adjatey Tetteh.

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
