## [Decision Letter · Decision Letter 0]

4 Jan 2026

PONE-D-25-19123Basic School Teachers’ Artificial Intelligence Literacy in Ghana: A Variance-Based Structural Equation Modelling ApproachPLOS One

Dear Dr. Arthur,

Thank you for submitting your manuscript to PLOS ONE. After careful consideration, we feel that it has merit but does not fully meet PLOS ONE’s publication criteria as it currently stands. Therefore, we invite you to submit a revised version of the manuscript that addresses the points raised during the review process.

We look forward to receiving your revised manuscript.

Kind regards,

Amir Karimi, PhD

Academic Editor

PLOS One

**Journal Requirements:**

2. In the online submission form, you indicated that “Data will be made available on reasonable request.”

4. Please ensure that you refer to Figure 3 in your text as, if accepted, production will need this reference to link the reader to the figure.

5. We note you have included a table to which you do not refer in the text of your manuscript. Please ensure that you refer to Table 4 in your text; if accepted, production will need this reference to link the reader to the Table.

Reviewers' comments:

Reviewer's Responses to Questions

**Comments to the Author**

1. Is the manuscript technically sound, and do the data support the conclusions?

Reviewer #1: Partly

Reviewer #2: Partly

2. Has the statistical analysis been performed appropriately and rigorously? 

Reviewer #1: Yes

Reviewer #2: Yes

3. Have the authors made all data underlying the findings in their manuscript fully available?

The PLOS Data policy requires authors to make all data underlying the findings described in their manuscript fully available without restriction, with rare exception (please refer to the Data Availability Statement in the manuscript PDF file). The data should be provided as part of the manuscript or its supporting information, or deposited to a public repository. For example, in addition to summary statistics, the data points behind means, medians and variance measures should be available. If there are restrictions on publicly sharing data—e.g. participant privacy or use of data from a third party—those must be specified.requires authors to make all data underlying the findings described in their manuscript fully available without restriction, with rare exception (please refer to the Data Availability Statement in the manuscript PDF file). The data should be provided as part of the manuscript or its supporting information, or deposited to a public repository. For example, in addition to summary statistics, the data points behind means, medians and variance measures should be available. If there are restrictions on publicly sharing data—e.g. participant privacy or use of data from a third party—those must be specified.requires authors to make all data underlying the findings described in their manuscript fully available without restriction, with rare exception (please refer to the Data Availability Statement in the manuscript PDF file). The data should be provided as part of the manuscript or its supporting information, or deposited to a public repository. For example, in addition to summary statistics, the data points behind means, medians and variance measures should be available. If there are restrictions on publicly sharing data—e.g. participant privacy or use of data from a third party—those must be specified.requires authors to make all data underlying the findings described in their manuscript fully available without restriction, with rare exception (please refer to the Data Availability Statement in the manuscript PDF file). The data should be provided as part of the manuscript or its supporting information, or deposited to a public repository. For example, in addition to summary statistics, the data points behind means, medians and variance measures should be available. If there are restrictions on publicly sharing data—e.g. participant privacy or use of data from a third party—those must be specified.

Reviewer #1: No

Reviewer #2: No

4. Is the manuscript presented in an intelligible fashion and written in standard English?

Reviewer #1: Yes

Reviewer #2: No

5. Review Comments to the Author

Reviewer #1: The manuscript presents a relevant study of AI literacy among basic school teachers in Ghana using a robust variance-based SEM approach. The main findings are novel and the statistical analysis is appropriate. However, substantial revisions are needed before publication.

Major comments:

1) Please streamline the introduction and literature review, reducing repetition and focusing on the main research gaps and rationale.

2) Expand on sampling criteria, potential sources of bias (e.g. only teachers aware of AI), and discuss how this may affect generalizability.

3) Clearly present all model limitations, ensuring interpretation of nonsignificant paths and overall study constraints is balanced and cautious.

4) Revise the Data Availability Statement to comply with PLOS ONE policy: data must be fully accessible through a public repository or similar resource.

5) Integrate the Ethics, Funding, and Conflict of Interest disclosures appropriately within the manuscript sections.

Minor comments:

1) Clarify some ambiguous sentences, especially in the discussion.

2) Check consistency of terminology and acronyms.

3) Improve structure: aim for concise paragraphs and clear transitions.

4) Ensure references are formatted according to PLOS ONE guidelines.

Overall, the work is scientifically interesting and potentially suitable for publication after major revision addressing the above points.

Reviewer #2: The manuscript addresses an important and timely topic concerning AI literacy among basic school teachers, and the study employs an appropriate methodological approach using PLS-SEM. However, several substantial revisions are required to enhance the scientific rigor and clarity of the work. First, although the statistical analysis is generally appropriate, it would benefit from deeper interpretation and additional analytical validation to strengthen the robustness of the findings. Second, the manuscript does not fully comply with the PLOS data availability policy, as the data are not openly accessible and are instead available only upon request, which is insufficient for publication. Third, while the paper is understandable, the writing requires considerable language editing to correct grammatical inconsistencies, reduce repetition, and improve the overall clarity and flow of ideas. Additionally, several core sections—such as the introduction, theoretical framework, discussion, and methodological justification—would benefit from tighter organization, stronger linkage to prior literature, and clearer articulation of the research gap and contributions. Ethical considerations are addressed but could be elaborated further to show stronger adherence to international standards. Overall, the manuscript has potential but requires major revisions to address methodological, structural, linguistic, and policy-related issues before it can be considered for publication.

6. PLOS authors have the option to publish the peer review history of their article (what does this mean?). If published, this will include your full peer review and any attached files.). If published, this will include your full peer review and any attached files.). If published, this will include your full peer review and any attached files.). If published, this will include your full peer review and any attached files.

...

Reviewer #1: **Yes:**Javid AbbasliJavid AbbasliJavid AbbasliJavid Abbasli

Reviewer #2: No

---

## [Author Response · Author response to Decision Letter 1]

25 Mar 2026

RESPONSE TO REVIEWERS’ COMMENTS

Manuscript ID: PONE-D-25-19123

Topic: Basic School Teachers’ Artificial Intelligence Literacy in Ghana: A Variance-Based Structural Equation Modelling Approach

Refined Topic (Based on the Reviewer’s recommendation): AI Literacy among Basic School Teachers in Ghana: A Structural Equation Modelling Analysis

Journal: PLOS One

Dear Editor,

We would like to thank you and the reviewers for reviewing this manuscript. Please, we have responded to the comments as follows:

Journal Requirements:

Response: Thank you for this comment. Please, we have rectified it as follows:

AI Literacy among Basic School Teachers in Ghana: A Structural Equation Modelling Analysis

Valentina Arkorful1, Francis Arthur2*, Ernest Opoku3, Ayishatu Ameen4, Iddrisu Salifu5, Sharon Abam Nortey2, Solomon Adjatey Tetteh2

1 College of Distance Education, University of Cape Coast, Cape Coast, Ghana

2 Department of Business and Social Sciences Education, Faculty of Humanities and Social Sciences Education, University of Cape Coast, Cape Coast, Ghana

3 Institute of Education, School of Educational Development and Outreach, College of Education Studies

4 Department of Agricultural Economics and Extension, School of Agriculture, University of Cape Coast, Cape Coast, Ghana

5 Centre for Coastal Management- Africa Centre of Excellence in Coastal Resilience, Department of Fisheries and Aquatic Sciences, University of Cape Coast, Cape Coast, Ghana

* Corresponding author

E-mail: a.francis1608@gmail.com/ francis.arthur007@stu.ucc.edu.gh (FA)

2. In the online submission form, you indicated that “Data will be made available on reasonable request.”

Response: Thank you for this comment. Please, we have indicated that:

Availability of data and materials

The datasets generated and/or analysed during the current study are not publicly available due to ethical and legal restrictions imposed by the University of Cape Coast (UCC) Institutional Review Board (IRB), which approved the study protocol. The data contain potentially sensitive information from human participants, including responses that could indirectly identify individuals or reveal personal and professional characteristics of basic school teachers. Public deposition of such data would therefore violate the conditions under which ethical approval was granted, as well as participants’ informed consent agreements guaranteeing confidentiality and anonymity.

However, the data can be made available upon reasonable request for academic and non-commercial research purposes, subject to approval and in compliance with applicable ethical guidelines. Requests for data access should be directed to the Institutional Review Board (IRB), University of Cape Coast, via the official contact: irb@ucc.edu.gh. The IRB serves as an independent, non-author body responsible for reviewing and approving data access requests to ensure adherence to ethical standards and participant confidentiality.

Response: Thank you for this comment. Please, have deleted the section which was captioned “Ethics approval” under the “declarations”.

4. Please ensure that you refer to Figure 3 in your text as, if accepted, production will need this reference to link the reader to the figure.

Response: Thank you for this comment. Please, we have stated that:

Figure 3 shows the bootstrapping results from PLS-SEM.

(see “Evaluation of Structural Model” section of the manuscript)

5. We note you have included a table to which you do not refer in the text of your manuscript. Please ensure that you refer to Table 4 in your text; if accepted, production will need this reference to link the reader to the Table.

Response: Thank you for this comment. Please, we have referred to Table 4 in the text. (see “Discriminant validity” section of the manuscript)

Response: Thank you for this comment.

Reviewers' comments:

Reviewer's Responses to Questions

Comments to the Author

1. Is the manuscript technically sound, and do the data support the conclusions?

Reviewer #1: Partly

Reviewer #2: Partly

Response: Thank you for the comments. Please, the manuscript is technically sound and also, the data support the conclusions of the study.

2. Has the statistical analysis been performed appropriately and rigorously?

Reviewer #1: Yes

Reviewer #2: Yes

Response: Thank you for this comment.

3. Have the authors made all data underlying the findings in their manuscript fully available?

Reviewer #1: No

Reviewer #2: No

Response: Dear Reviewer, thank you for the comments. Please, we indicated that:

Availability of data and materials

The datasets generated and/or analysed during the current study are not publicly available due to ethical and legal restrictions imposed by the University of Cape Coast (UCC) Institutional Review Board (IRB), which approved the study protocol. The data contain potentially sensitive information from human participants, including responses that could indirectly identify individuals or reveal personal and professional characteristics of basic school teachers. Public deposition of such data would therefore violate the conditions under which ethical approval was granted, as well as participants’ informed consent agreements guaranteeing confidentiality and anonymity.

However, the data can be made available upon reasonable request for academic and non-commercial research purposes, subject to approval and in compliance with applicable ethical guidelines. Requests for data access should be directed to the Institutional Review Board (IRB), University of Cape Coast, via the official contact: irb@ucc.edu.gh. The IRB serves as an independent, non-author body responsible for reviewing and approving data access requests to ensure adherence to ethical standards and participant confidentiality.

4. Is the manuscript presented in an intelligible fashion and written in standard English?

Reviewer #1: Yes

Reviewer #2: No

Response: Thank you for the comments. Please, the manuscript has been presented in an intelligible fashion and written in standard English.

5. Review Comments to the Author

Reviewer #1: The manuscript presents a relevant study of AI literacy among basic school teachers in Ghana using a robust variance-based SEM approach. The main findings are novel and the statistical analysis is appropriate. However, substantial revisions are needed before publication.

Response: Dear Reviewer, thank you for the comments. Please, we have addressed the comments below.

Major comments:

1) Please streamline the introduction and literature review, reducing repetition and focusing on the main research gaps and rationale.

Response: Dear Reviewer, thank you for the comments. Please, we have streamline and introduction and literature review. (see “Revised version of the manuscript).

2) Expand on sampling criteria, potential sources of bias (e.g. only teachers aware of AI), and discuss how this may affect generalizability.

Response: Dear Reviewer, thank you for the comments. Please, we indicated that:

Teachers were purposively selected based on their prior awareness of AI, which, while ensuring informed responses, may introduce selection bias and limit the generalizability of the findings to the broader population of basic school teachers in Ghana who may have limited or no exposure to AI.

(see “Study design an population” section of the revised manuscript)

3) Clearly present all model limitations, ensuring interpretation of nonsignificant paths and overall study constraints is balanced and cautious.

Response: Dear Reviewer, thank you for the comments. Please, we interpreted all the non-significant paths. (see “Evaluation of structural model” section of the revised manuscript).

4) Revise the Data Availability Statement to comply with PLOS ONE policy: data must be fully accessible through a public repository or similar resource.

Response: Dear Reviewer, thank you for the comments. Please, we indicated that:

Availability of data and materials

The datasets generated and/or analysed during the current study are not publicly available due to ethical and legal restrictions imposed by the University of Cape Coast (UCC) Institutional Review Board (IRB), which approved the study protocol. The data contain potentially sensitive information from human participants, including responses that could indirectly identify individuals or reveal personal and professional characteristics of basic school teachers. Public deposition of such data would therefore violate the conditions under which ethical approval was granted, as well as participants’ informed consent agreements guaranteeing confidentiality and anonymity.

However, the data can be made available upon reasonable request for academic and non-commercial research purposes, subject to approval and in compliance with applicable ethical guidelines. Requests for data access should be directed to the Institutional Review Board (IRB), University of Cape Coast, via the official contact: irb@ucc.edu.gh. The IRB serves as an independent, non-author body responsible for reviewing and approving data access requests to ensure adherence to ethical standards and participant confidentiality.

(see “Availability of data and materials” section)

5) Integrate the Ethics, Funding, and Conflict of Interest disclosures appropriately within the manuscript sections.

Response: Dear Reviewer, thank you for the comments. Please, we indicated that:

Funding

The authors have confirmed that they did not receive any financial support for the research, authorship, and publication of this article.

Competing interests

The authors declare that they have no competing interests.

Also, the Ethics part of the study is at the materials and methods section. This was made so based on the Editor’s recommendation. Thank you

We indicated that:

Ethical considerations

The study prioritised the ethical treatment of participants. Ethical approval for the study was granted by the Institutional Review Board of the University of Cape Coast (ID: UCCIRB/CES/2023/169). Following the approval, data collection took place from March 04, 2024, to July 26, 2024. All participants were made to sign written informed consent before taking part, and the researchers guaranteed the confidentiality and anonymity of their responses. In accordance with ethical guidelines, participants were free to withdraw from the study at any time without consequence. To ensure data security and privacy, information was stored securely and only the research team had access to it.

Minor comments:

1) Clarify some ambiguous sentences, especially in the discussion.

Response: Dear Reviewer, thank you for the comments. Please, we have rectified all ambiguous sentences, especially in the discussion.

2) Check consistency of terminology and acronyms.

Response: Dear Reviewer, thank you for the comments. Please, we have checked and rectified any inconsistency issues in terms of terminology and acronyms.

3) Improve structure: aim for concise paragraphs and clear transitions.

Response: Dear Reviewer, thank you for the comments. Please, we have improved the structure of the manuscript.

4) Ensure references are formatted according to PLOS ONE guidelines.

Response: Dear Reviewer, thank you for the comments. Please, we have formatted the references to be in line with PLOS ONE guidelines.

Overall, the work is scientifically interesting and potentially suitable for publication after major revision addressing the above points.

Response: Dear Reviewer, thank you for the comments. Please, we revised the manuscript based on the suggestions. Thank you

Reviewer #2: The manuscript addresses an important and timely topic concerning AI literacy among basic school teachers, and the study employs an appropriate methodological approach using PLS-SEM. However, several substantial revisions are required to enhance the scientific rigor and clarity of the work.

First, although the statistical analysis is generally appropriate, it would benefit from deeper interpretation and additional analytical validation to strengthen the robustness of the findings.

Response: Dear Reviewer, thank you for the comments. Please, the statistical analysis has been well interpreted and stated. (see “Results” section of the manuscript, thank you)

Second, the manuscript does not fully comply with the PLOS data availability policy, as the data are not openly accessible and are instead available only upon request, which is insufficient for publication.

Response: Dear Reviewer, thank you for the comments. Please, we have indicated that:

Availability of data and materials

The datasets generated and/or analysed during the current study are not publicly available due to ethical and legal restrictions imposed by the University of Cape Coast (UCC) Institutional Review Board (IRB), which approved the study protocol. The data contain potentially sensitive information from human participants, including responses that could indirectly identify individuals or reveal personal and professional characteristics of basic school teachers. Public d

---

## [Editor Report · Decision Letter 1]

29 Mar 2026

AI Literacy among Basic School Teachers in Ghana: A Structural Equation Modelling Analysis

PONE-D-25-19123R1

Dear Dr. Arthur,

We’re pleased to inform you that your manuscript has been judged scientifically suitable for publication and will be formally accepted for publication once it meets all outstanding technical requirements.

Kind regards,

Amir Karimi, PhD

Academic Editor

PLOS One
---

## [Editor Report · Acceptance letter]

PONE-D-25-19123R1

PLOS One

Dear Dr. Arthur,

I'm pleased to inform you that your manuscript has been deemed suitable for publication in PLOS One. Congratulations! Your manuscript is now being handed over to our production team.

Kind regards,

on behalf of

Dr. Amir Karimi

Academic Editor

PLOS One